# Digitalisation and Economic Growth in the European Union

**Petru Ovidiu Mura * and Liliana Eva Donath** 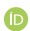

Department of Finance, Faculty of Economics and Business Administration, West University of Timişoara Romania, 300223 Timişoara, Romania; liliana.donath@e-uvt.ro
* Correspondence: petru.mura@e-uvt.ro

**Abstract:** The aim of the present paper is to analyse the effects of digitalisation on economic growth in the European Union. An econometric model with balanced panel data is used, with the analysis spanning over a 22-year time frame from 2000 to 2021. The main conclusion is that digitalisation generates a positive and significant impact on economic growth, even when several control variables are taken into consideration. The results prove their robustness, which is backed by the employment of the DESI as an independent variable. This paper contributes to the existing empirical analyses by extending the research on digitalisation to the entire EU, and separately for the old EU-15 member states and EU-13 new member states since, to our best knowledge, the existing literature has not approached the subject in this manner. As a policy recommendation, we suggest that public decision-makers take measures that support more harmonised digitalisation policies, favouring the new business model based on digitalisation.

**Keywords:** digitalisation; economic growth; panel model; fixed and random effects; European Union

## 1. Introduction

The transition from the traditional business model to the digitalised one raises several questions and concerns regarding the challenges that modern industries will meet from economic, technological, financial, and social stances. Since it is a dynamic, ongoing process, the answers are not straightforward. In addition, industries, regions, and countries are at different stages of digitalisation; developed countries are the most advanced while others countries that are less developed are lagging. Therefore, interruptions and blockages may occur in the formation of business networks impeding digitalisation to reveal its full potential for economic growth. Nevertheless, the digitalisation process has intensified over the last decades posing a wide range of research questions at macro and microeconomic levels.

Digitalisation is a concept of many parts. According to context, it is referred to from technical, economic, financial, and social perspectives. It includes, among others, communication and information connectivity, AI, the internet of things, platform and blockchain technologies, virtual reality, etc. According to a widely accepted definition, it relates to the digital transformation of the economy, which is regarded as a new approach for businesses based on innovation [1]. The core significance that is given to digitalisation is its aim to change the business model and provide added value. As it is stated in the literature, industry 4.0 requires new practices that *"include collecting and sharing data using sensors in interconnected devices and using digital technologies for processing and analysing data to take new actions". Consequently, digitalisation is considered "the process of adopting such data-capturing and analysis technologies"* [2].

Hence, the promotion of digitalisation on a large scale represents the foundation of changes that occur in the existing economic growth pattern, which is based on traditional industrial structures. It enhances the role of innovation-based technologies and big data analyses in productivity growth and, ultimately, helps with the completion of sustainable development goals.

The acceleration of the digital economy also triggers structural changes in business, governmental, and institutional fields, including transformations in economic, financial, and social behaviours. It demands a heterodox, out-of-the-box approach, requires the intervention of multiple stakeholders [3], (i.e., businesses, customers, suppliers, industry networks, etc.) reorients resources, promotes accessibility, and unleashes creativity. Last but not least, digitalisation raises huge research opportunities, given its disruptive effect on the economy over the past few decades [4].

Undoubtedly, numerous factors contribute to the successful implementation of digitalisation, the institutional ones being the main drivers of systemic and harmonising principles that are applicable in the EU's countries and regions [5].

One frequently debated issue is the impact of digitalisation on productivity and economic growth, respectively. Lato sensu, it is reasonable to assume that digitalisation has a significant positive macroeconomic impact. Nevertheless, in-depth studies are needed to depict the influence of different determinants and eliminate possible blockages.

The aim of the paper is to find, in a comprehensive manner, to what extent the EU-15 and the EU-13 groups of countries experience the same stance on digitalisation and whether the hypothesis is verified for both groups The conclusions may serve as policy lessons that are needed to harmonise productivity and competitiveness in the EU.

The research hypothesis of the present paper is that digitalisation positively impacts economic growth in the EU-28.

The hypothesis is verified by the following two stages:

A. In the first stage, we employ selected variables that proxy the digitalisation infrastructure (i.e., fixed broadband subscriptions/100 inhabitants, mobile phone subscriptions/100 inhabitants, and the percentage of internet users). These proxies are in line with [6–9] and were used since they are indispensable prerequisites of digitalisation, and they show the degree of penetration of the new communication technologies in the economy and society. These proxies span two decades (from 2000 to 2021) to investigate whether the correlation holds over a long time frame. If, at the beginning of the 2000s, gains following digitalisation were modest, over the considered time frame, it has already become an indispensable tool for productivity gains and economic growth, as will be further demonstrated.

B. In the second stage of the study, we include the DESI (The Digital Economy and Society Index) as an independent variable for a more in-depth view regarding the effects of digitalisation, in its entirety, on economic growth. In this case, the analysis spans 5 years (2017–2021) according to the availability of data. The DESI captures the advances and performances made by the EU member states in the field of digitalisation. The UK is not included in database.

Consequently, from the perspective of the present research, digitalisation is regarded in a comprehensive manner by including the infrastructure assets (indispensable to ensure connectivity), e-commerce, e-business, e-government technologies, software assets, and intangibles (e.g., digital skills) as derived from the components of the DESI.

In terms of our research method, a panel data approach is used. As control variables, foreign direct investment, trade openness, population growth rate, gross fixed capital formation, government consumption expenditure, public debt, and recycling rate of municipal waste were used.

The article concludes that the correlation between the dependent variable (GDP per capita) and the independent variable (digitalisation) is positive in both groups of countries (i.e., EU-15 and EU-13). The conclusion is endorsed following the employment of the DESI.

The research contributes to the existing literature by shedding light on the possible reasons that explain the results in the EU in its entirety, thus filling a gap since the vast majority of the literature emphasises the relevance of business and/or industry-related digitalisation, country-specific stances, or refers to its impact on regional development. It also highlights the similarities and differences between the EU-15 and EU-13 groups of countries.

The article proceeds as follows: literature review; some statistical considerations; data and methodology; results; discussion; and conclusions. The remainder of the paper is dedicated to research limitations.

## 2. Literature Review

There is a large amount of literature that studies the economic impact of the various components of digitalisation, which is one of the major issues concerning the effects on productivity [10]. The authors show the long-term positive effects on labour-saving costs as well as on productivity increases following the automation of Spanish companies. It is also stressed that whereas robotisation, for example, induces positive effects on productivity, it seems that digitalisation did not manifest the same effect, seemingly, because the e-commerce did not entirely transfer its impact. In the same line of thinking, ref. [11] provides evidence on the correlation between innovation and digitalisation, on the one hand, and productivity, on the other hand, in small- and medium-sized firms in South Africa. Nevertheless, there is a considerable difference in accessing digital technologies and large companies manifesting a comparative advantage. Moreover, there are persistent barriers in developing countries such as low digital skills, low-quality internet and electricity infrastructure, low access to funding for small companies, etc. To complement these findings, ref. [12] put forward the importance of innovation in linking digitalisation with companies' performance, finding that accessibility to digitalisation and the extent of digitalised operations are significant determinants. The literature often concludes that digitalisation leads to new business models that are boosted by the degree of innovation and digitalisation that enhances connectivity, know-how, flexibility, etc. [13]. Conducting a study for the automotive and media industries [1] demonstrates that there are differences concerning the extent that digitalisation contributes to innovation, admitting that there are still several challenges in observing the full benefits of digitalisation in adopting the new business model.

In an extensive research, ref. [14] assesses the importance of digitalisation at company level on productivity gains using cross-country data. Their study supports the general conclusion that digitalisation is a driver of higher productivity but not to the same degree for all companies. They also argue that individual digitalisation gains have weak aggregate influence because of different ratios to implement digital technologies and non-homogeneous digital skills. In addition, several authors [15] discuss the impact of the internet of things on a wide range of manufacturing companies, highlighting the concerning issues for the considered fields, from qualified staff to suppliers.

In [16], the author studies the involvement of new technologies with traditional ones, stressing the positive impact of digitalisation, which calls for new business strategies.

Acknowledging the channels through which digital technologies [17] impact productivity, it is argued that productivity mismeasurement underestimates the effects of digitalisation, mainly on the smaller components as well as the present structure of GDP measurements that do not include the free services provided by the internet.

In their extensive research [18], the authors bring forward the importance of intangibles and digitalisation in Dutch firms, demonstrating the sector-wide advantage of companies that enhance the digital skills that prove beneficial for productivity growth. They also prove that with investments in digital technologies and using the potential of intangibles (i.e., digital skills), low-productivity companies can catch up on productivity that is eventually reflected at the aggregate level.

Other authors [19] stress the importance of managerial decisions to support companies' development, emphasising the role of accurate definitions of digitalisation. Discussing the level of digitalisation [3], the results show significant differences among countries calling for more governmental policy interventions in R&D, mainly in countries that lag and do not use their entire capabilities to innovate. The authors also find that there is a weak or insignificant correlation between the digitalisation index for large enterprises

and GDP/capita. A strong correlation was, nevertheless, found in the case of medium-size enterprises.

Another strand of the literature refers to the response of economic growth to digitalisation. Over the last 30 years, numerous scientific works [20,21] have debated and researched this topic. In [22], the author argues that the effects of digitalisation are not completely noticeable because AI and IT innovations are not used on a sufficiently large scale. In addition, ref. [23] demonstrates that African countries have not yet fully benefited from digitalisation considering that specific skills, insufficient ICT infrastructure, etc., have contributed to this stance despite the efforts that have been made.

Other articles [24] look into the AI effects on economic growth by summarising scholars' contributions who use neoclassical, task-based, or empirical models. The authors conclude that AI is at its earliest stage yet; hence, uncertainties are raised concerning its influence on economic growth. Being aware of the impact of digitalisation on sustainable development [25], the authors show that environmentally friendly technologies ensure the preservation of information and its rapid distribution in any location it may be needed. Going more in-depth, ref. [26] refers to the possibility to ensure economic growth while environmental degradation is stopped using digital technology. On the other hand, digital technologies may be harmful to the biosphere, and, therefore, the production process should be redesigned to meet the sustainability requirements.

Other authors [27] consider the positive effects of digitalisation during pandemics, which hugely increased the demand for digital technology. Despite this reality, referring to the "belt and road" countries, they conclude that, at least for these countries, the correlation between digitalisation and economic growth remains rather ambiguous, calling for more integration of digitalisation and the real economy as well as more cooperation in the region.

Digitalisation also transforms views on competition and economic growth, according to [28]. Using a set of relevant indicators and indexes, the authors argue that, to support competition, more is needed given the imbalances among countries regarding the implementation of ICT. Ref. [29] also stresses the need for policies that support digitalisation given the specificities of geographical regions.

Studies [30] also provide explanations concerning the weak impact of AI on economic growth. The authors show that, unless productivity gains are reflected in individual incomes, aggregate demand stagnates and impedes growth. It is suggested that more research is needed on the effects of AI, mainly on labour and production. In this line of thought, ref. [31] refers to the mechanisms of the diffusion of digitalisation and how institutions and governance are affected. Apparently, in higher quality-driven institutions, digitalisation is faster. Because the implementation of digitalisation is not homogenous among countries and groups of countries, its impact on the real GDP per capita should be studied individually. In a paper dedicated to the importance of the public sector, [31] stresses the factors that induce some drawbacks in the digitalisation process in Russia. The role of the public sector is also emphasised as the main driver of accelerating the digitalisation process by implementing an open information space system.

This present paper complements the existing literature by bringing forward the effect of digitalisation on economic growth in the EU-28 and seeks to find the similarities and differences between the EU-15 and the EU-13 groups of countries.

## 3. Some Statistical Considerations

The statistical background displays the relevant information concerning the digital stance in EU countries.

Figure 1 illustrates the differences in economic development and labour productivity in the EU, showing rather large discrepancies among countries. While Luxemburg records the highest GPD per capita, Bulgaria and Romania lag. This lack of homogeneity in productivity and economic development is expected to influence the degree of digitalisation across the EU and the overall competitiveness of the region.

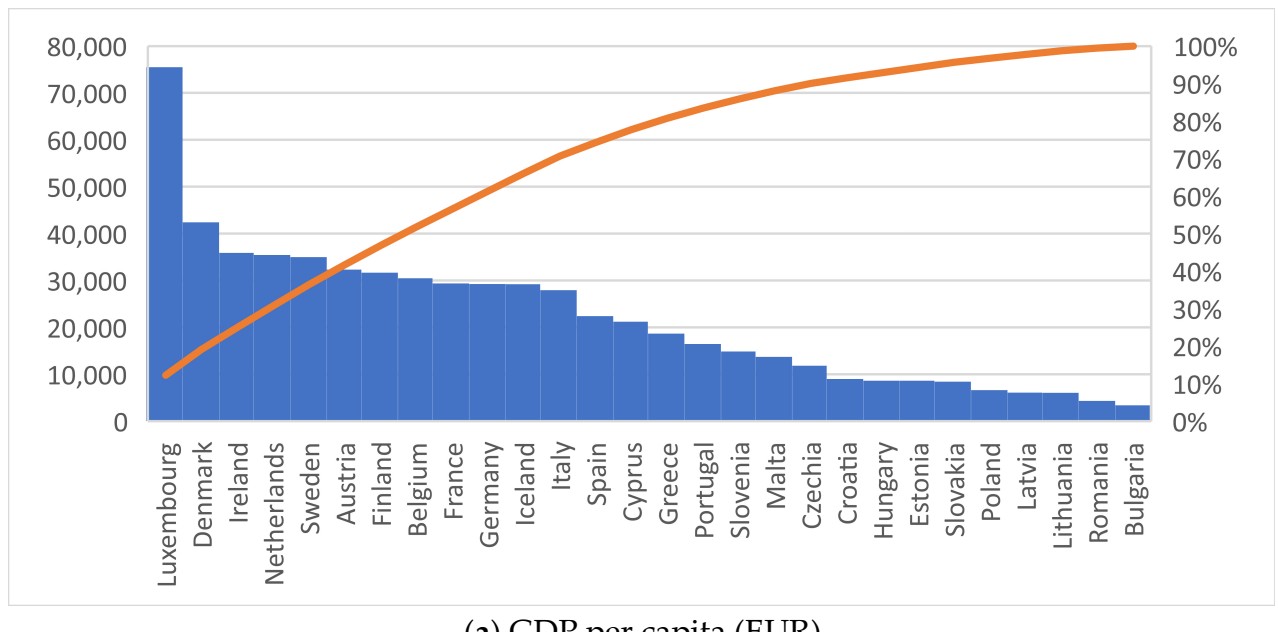

(**a**) GDP per capita (EUR)

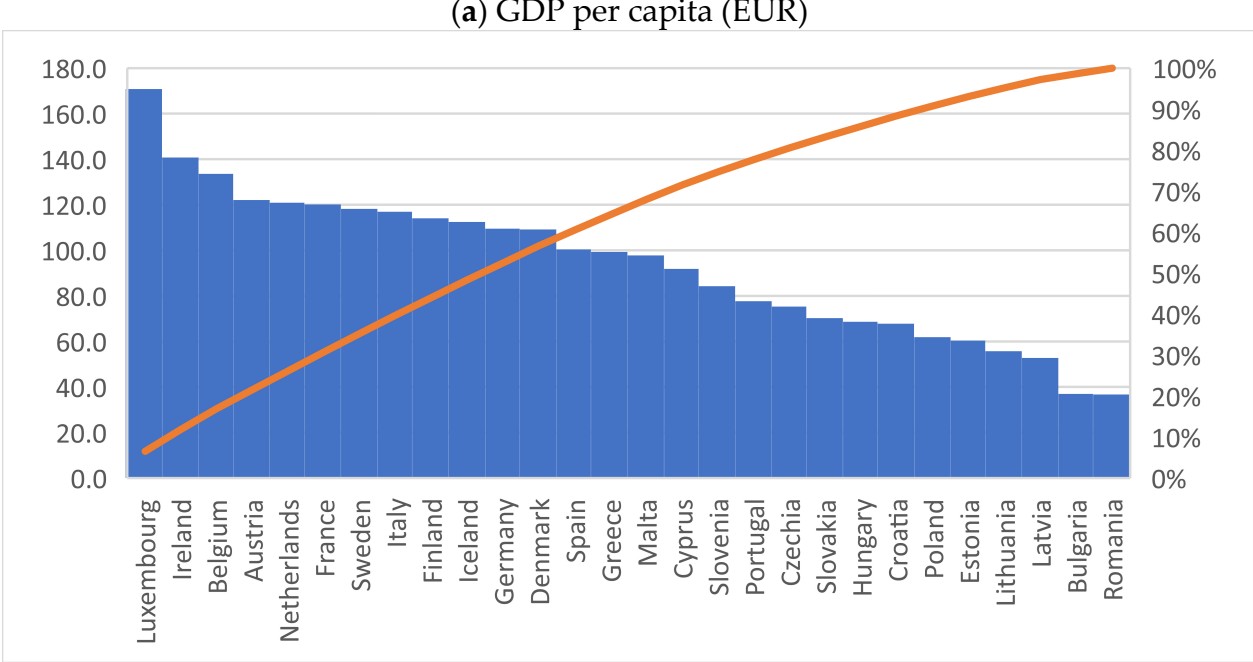

(**b**) Labour productivity (% of the EU)

**Figure 1.** The real GDP per capita and labour productivity in the EU (source: Eurostat).

According to the Eurostat database [32], there is a lack of homogeneity among the EU countries regarding digitalisation. As Figure 2a shows, Romania leads, recording the highest digital intensity, but, paradoxically, it records the lowest labour productivity, meaning that the digital technologies are concentrated, at large, in urban-located companies, whereas other, smaller companies lack such investments. Croatia has the highest percentage of enterprises linked with customers and suppliers. Austria is most advanced in using marketing-related software, and Portugal and Romania are most oriented towards the sustainable use of digital equipment. Nevertheless, concerning digital skills (Figure 3a), Romania lags, with Iceland recording the best skills, while in Estonia, individuals have carried out the most internet-related activities (Figure 3b).

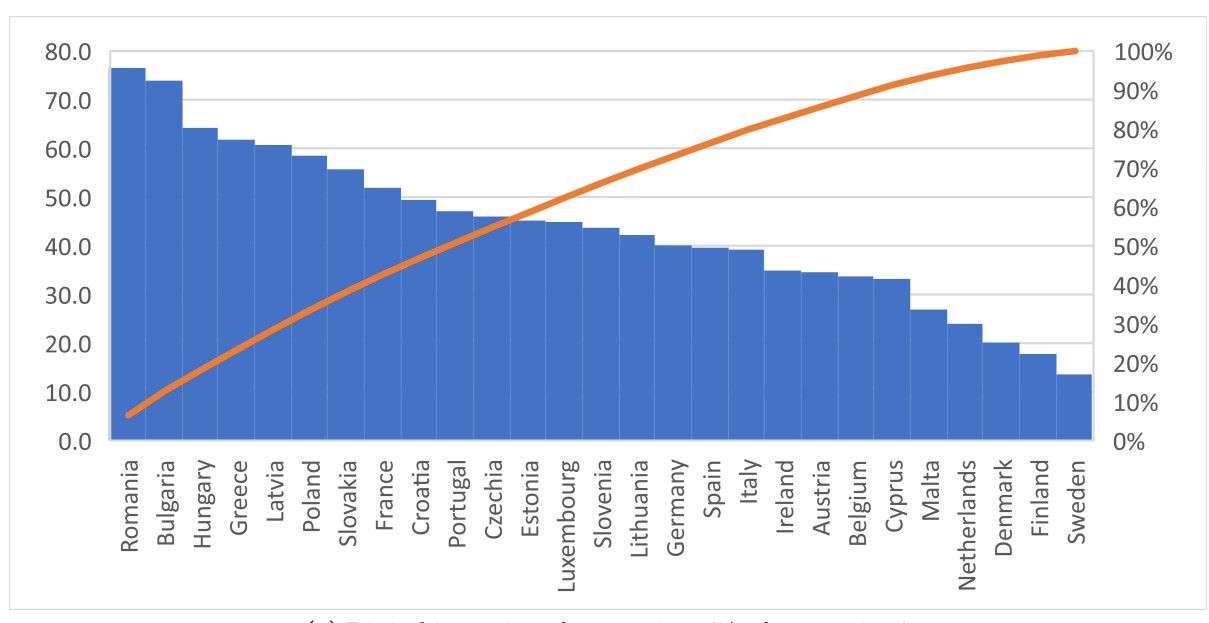

(**a**) Digital intensity of enterprises (% of enterprises)

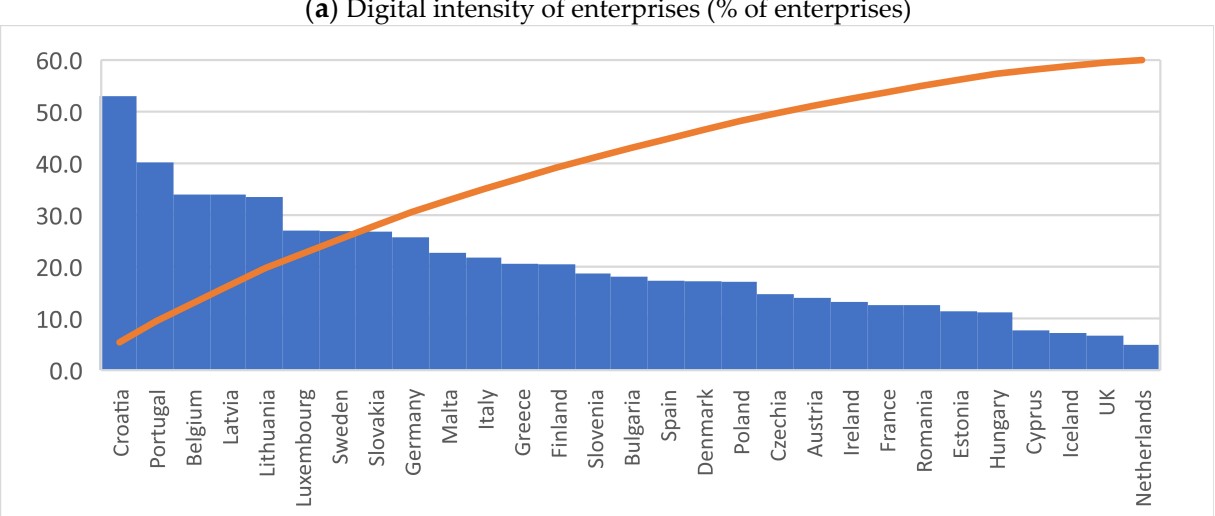

(**b**) Percentage of businesses automatically liked to customers/suppliers

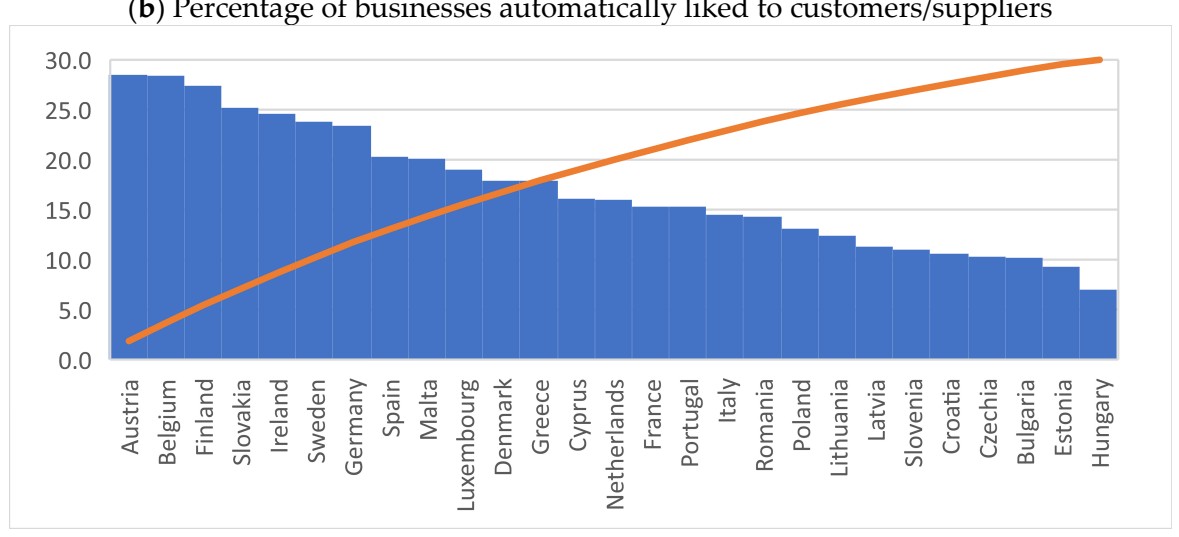

(**c**) Enterprises using software for marketing purposes (% of enterprises)

**Figure 2.** Selected aspects of digitalisation in businesses (source: Eurostat).

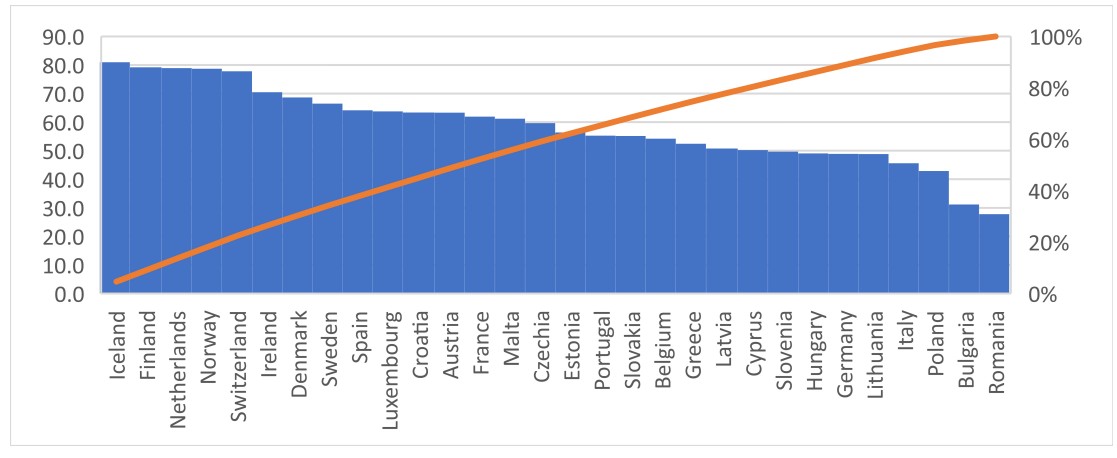

(**a**) Basic digital skills

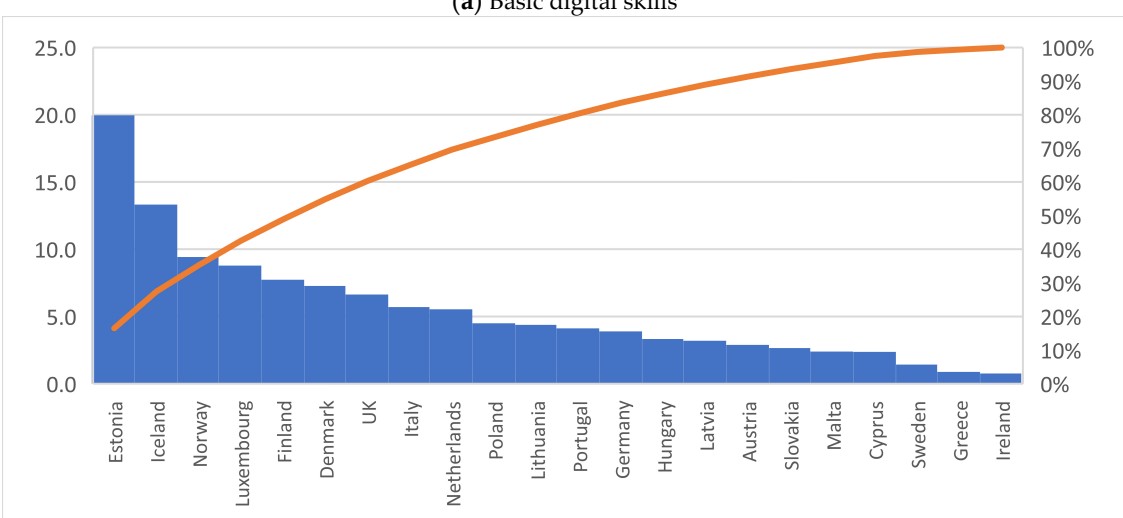

(**b**) Individuals who have carried out 5 or 6 of the related internet activities

**Figure 3.** Basic skills and internet-related activities (source: Eurostat).

Figure 4 reveals the differences between the EU countries regarding the DESI composite index. The highest level of the index is recorded by the Scandinavian countries while Romania is last. Nevertheless, all countries have improved their digital stance between 2017 and 2021.

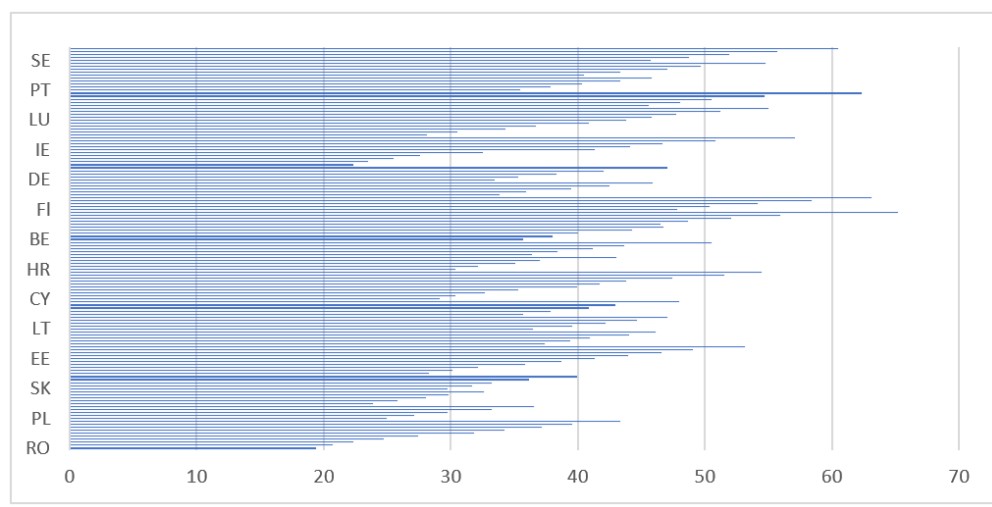

**Figure 4.** The DESI composite index in selected EU countries.

Although companies have accessed digital technology on a rather large scale or digitally interacted with their customers, according to a survey conducted by the ECB [33], digitalisation has been perceived as a shock, with companies also acknowledging the obstacles that prevent them from becoming fully digital, i.e., reorganising the business, recruitment of highly skilled staff, cost of ICT implementation, regulation and legislation, etc. On the same line, the OECD [34] has identified some deficiencies that explain why digitalisation gains have not been unrolled to all companies. One of these factors refers to the complementarities between digital technology as well as the company's technical and managerial skills, capabilities, assets, etc. In addition, new companies are more inclined to adopt digitalisation, while the traditional ones are, seemingly, not sufficiently equipped, lack the necessary capital and skills and remain rather reluctant in this respect. According to [35], the low labour-intensive, agriculture, basic goods manufacturing, education, health, and entertainment sectors are lagging.

## 4. Data and Research Methodology

### 4.1. Data and Variable

The impact of digitalisation on economic growth is based on a set of balanced data, with 28 cross-sections (EU-28 countries), for 2000–2021, using a panel model approach. A panel data approach is used because it supports the control for individual heterogeneity and gives more informative data, more variability, less collinearity among the variables, and more efficiency [36]. The selected countries are presented in Table 1. The annual data regarding the dependent, explanatory, and control variables were selected from the official available sources.

**Table 1.** List of analysed countries.

| Countries | | |
|---|---|---|
| Austria | Germany | Poland |
| Belgium | Greece | Portugal |
| Bulgaria | Hungary | Romania |
| Croatia | Ireland | Slovak Republic |
| Cyprus | Italy | Slovenia |
| Czech Republic | Latvia | Spain |
| Denmark | Lithuania | Sweden |
| Estonia | Luxembourg | United Kingdom |
| Finland | Malta | |
| France | Netherlands | |

Starting from the reviewed literature, the tested hypotheses are as follows:

**H0:** *Digitalisation positively impacts economic growth in EU-27 countries and the UK.*

**H1:** *Digitalisation has a negative or no impact on economic growth in EU-27 countries and the UK.*

The null hypothesis was suggested by the scatter graph of the function, presented in Figure 5. The "Regression Line" method suggests a positive linear connection between the chosen variables.

The first stage of analysis verifies the hypothesis and investigates, over two decades, the correlation between digital infrastructure (proxied by mobile subscriptions, internet users, and fixed broadband subscriptions) as a prerequisite for the digitalisation process. In the model, economic growth is considered an independent variable, proxied by the GDP per capita, which is largely used in the literature as a statistical tool to measure a country's overall level of economic performance. GDP per capita is computed as the ratio of the GDP to the average population quantified in purchasing power standards (PPS, EU-28—2020). The interest variable (i.e., digitalisation) is proxied by mobile subscriptions, internet users, and fixed broadband subscriptions, which serve as prerequisites for digitalisation and as potential levers for economic growth [6].

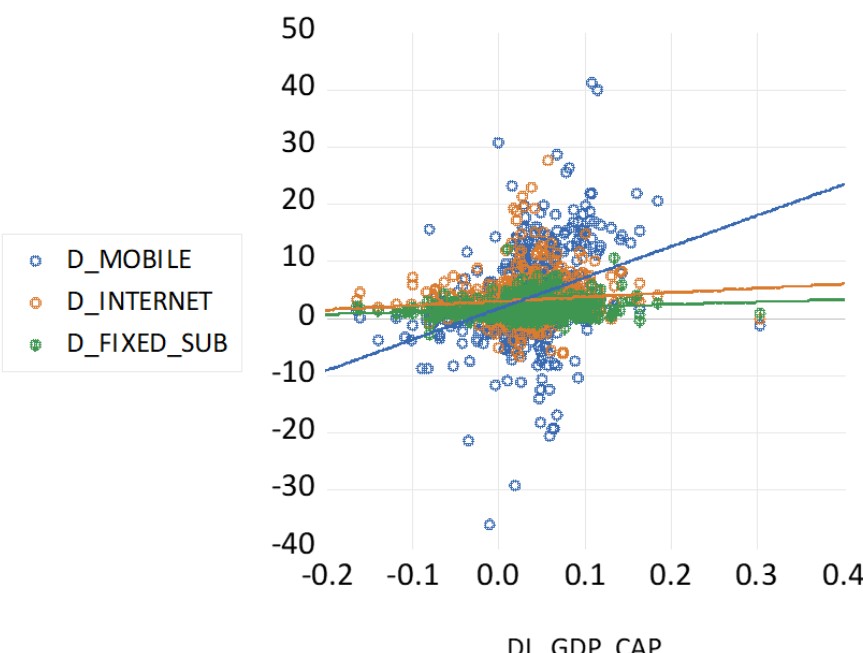

**Figure 5.** The relationship between digitalisation variables and economic growth. Notes: D_MOBILE, D_INTERNET, and D_FIXED_SUB refer to our interest variables, which are stationary in first difference; DL_GDP_CAP is the logarithm of the dependent variable, which is stationary in first difference also.

In the second stage of the analysis, these findings are complemented by introducing the DESI as an independent variable [37], but for a shorter, 5-year time frame (2017–2021), according to the data availability.

The above-mentioned digital infrastructure-related proxies are fundamental (among others) in supporting the digitalisation process since they allow fast and transparent access to information, innovation, real-time communication, e-business, e-commerce, e-governance, etc. Henceforth, they are susceptible to enhancing productivity, efficiency, and profitability.

The DESI is a composite indicator calculated by the European Commission that illustrates the progress made by the member states on their path towards digitalisation. The DESI embeds four dimensions related to human capital, connectivity, integration of digital technology, and digital public services [37]. Each dimension is subdivided to better capture as many digitalisation aspects as possible. It should be noted that the infrastructure components as defined in our research are included in the connectivity dimension; the human capital dimension includes digital skills, integration of digital technology, the digital intensity of business (AI, cloud, big data, etc.), as well as e-commerce and business-related technology, whereas digital public services include e-governance aspects.

According to the International Telecommunications Union, mobile-cellular telephone subscriptions refer to the number of subscriptions to a public mobile-telephone service that provides access to the PSTN (public switched telephone network) using cellular technology. The second variable, internet users, refers to the individuals who use the internet, following a survey addressed to national households. It is measured as a percentage of the total population of a country. Finally, fixed-broadband subscriptions refer to fixed subscriptions to high-speed access to the public internet; they include both residential as well as organisational subscriptions, calculated as fixed broadband subscribers/population. The number of internet users and broad subscribers shows the propensity to use new technology as well as the access to the basic communication infrastructure that may explain the level of digital skills and digitalisation in a country.

To overcome omitted variable bias in the model, as well as to isolate the reaction of the interest variables, a set of control variables is considered for economic growth, as

follows: foreign direct investment, trade openness, population growth rate, gross fixed capital formation, government consumption expenditure, public debt, and recycle rate of municipal waste.

Foreign direct investments denominated in current U.S. dollars show the net outflows of investment from the reporting economy to the rest of the world as a percentage of the GDP. Many studies have endeavoured to find whether FDIs impact economic growth. The results have been mixed, but most authors argue that FDIs contribute to economic growth. Ref. [38] investigates the relationship between FDI and economic growth in North African countries, finding a positive and strongly significant relationship. Ref. [39] investigates the relationship between FDIs and economic growth in 16 Arab countries between 1970 and 2008, strengthening the opinion that FDIs manifest a positive and significant impact on economic growth.

Trade openness measures the degree of openness of a country to international commerce. It is quantified as the percentage of imports and exports of goods and services in the GDP. Trade openness impacts growth by increasing income per capita by boosting productivity through competition. Openness supports technological transfer across borders and, hence, a more efficient organisation of the economy. Consequently, a positive sign for this variable is expected.

The population growth rate is the exponential rate of increase in the midyear population from year t−1 to t. It is a significant factor of economic growth, and therefore, we expect a negative coefficient since a lower population growth rate usually relates to higher GDP per capita. We appreciate that population growth is also determinant of the spread of digitalisation (via telecommunication) and its relation to economic growth [40].

Gross fixed capital formation (i.e., gross domestic investment) as a percentage of GDP includes, according to the World Bank, several assets such as land improvements, plant, machinery, equipment purchases, and the construction of roads and railways, including schools, offices, hospitals, private residential dwellings, and commercial and industrial buildings.

Government consumption expenditure, as a percentage of the GDP, shows the annual public purchases of goods and services. These two variables are significant growth factors, as government economic and social policies affect production factors, e.g., by financing education and infrastructure. It is expected that government investment has a positive sign, while for government expenditure, the expected sign is not straightforward, given the (un)productive nature of most of the public expenditure.

Public debt refers to the ratio of the government's consolidated gross debt to the GDP. The correlation between public debt and economic growth is addressed by [41], who finds a negative correlation between the two variables, in a sample of OECD countries. Ref. [42] examines how public debt affects economic growth by analysing the related literature, including different countries, income size, debt levels, and research methods. They conclude that the correlation can be either positive, negative, or non-linear.

The recycling rate of municipal waste (as a proxy for circular economy) measures the percentage of recycled waste in the generated municipal waste showing the propensity towards sustainability. It includes a wide range of material recycling, composting, and anaerobic digestion. Elaborating on the same topic, ref. [43] investigates the non-linear impact of municipal solid waste recycling and energy efficiency on environmental performance and economic growth in the USA. They argue that recycling contributes to the quality of the environment, economic growth, and energy efficiency. In addition, significant contributions regarding the relationship between municipal solid waste recycling and economic growth are provided by [44]. Referring to the USA, they employ bootstrapping autoregressive distributed lag modelling for investigating the cointegration relationship among MSW (municipal solid waste) recycling, economic growth, carbon emissions, and energy efficiency, using quarterly data from 1990 to 2017. They conclude that a positive uni-directional causality from MSW recycling to economic growth, carbon emissions, and energy efficiency exists.

The descriptive statistics of the variables, as well as detailed information in terms of content, the scale of measurement, the source of data, and their expected signs, are presented in Tables 2 and 3.

**Table 2.** Descriptive statistics of selected variables.

| | Mean | Median | Maximum | Minimum | Std. Dev. | Observations |
|---|---|---|---|---|---|---|
| GDP/CAP | 25,372.52 | 23,800 | 79,600 | 5300 | 11,539.3 | 575 |
| MOBILE | 110.76 | 114.62 | 172.15 | 16.77 | 25.77 | 575 |
| INTERNET | 65 | 70 | 98.86 | 4.53 | 22.63 | 575 |
| FIXED_SUB | 22.20 | 24.41 | 47.49 | 0.011 | 12.66 | 575 |
| DESI | 40.51 | 40.51 | 65.25 | 19.39 | 9.56 | 135 |
| FDI | 8.43 | 2.0004 | 300.40 | −87.22 | 33.93 | 575 |
| TRADE | 116.91 | 102.97 | 380.10 | 22.28 | 63.59 | 575 |
| GOV_EXP | 19.89 | 19.56 | 27.93 | 12.01 | 2.87 | 575 |
| GDI | 22.16 | 21.71 | 54.30 | 10.68 | 4.28 | 575 |
| POP_GR | 0.293 | 0.265 | 3.93 | −3.74 | 0.917 | 575 |
| PD | 58.94 | 53.6 | 186.4 | 3.8 | 34.25 | 575 |
| RECYCLE | 30.09 | 30.7 | 69.8 | 0 | 17.93 | 575 |

Source: own calculation using Eviews 13 software. Notes: the variables are defined as follows: GDP per capita (GDP/CAP), mobile subscriptions (MOBILE), internet users (INTERNET), fixed subscription (FIXED_SUB), foreign direct investment (FDI), trade openness (TRADE), government expenditures (GOV_EF), gross domestic investment (GDI), population growth rate (POP_GR), public debt (PD), and municipal recycling rate (RECYCLE).

**Table 3.** Description of variables and their expected sign.

| Variables | Explanation | u. m. | Source | Expected Sign |
|---|---|---|---|---|
| Economic growth—dependent variable (GDP/CAP) | GDP per capita | PPS | Eurostat database (2023) | |
| Interest variables (digitalisation) | Mobile subscriptions Internet users Fixed broadband subscriptions | % | International Telecommunications Union database (2023) | + |
| | DESI | | European Commission (2022) | + |
| **Controls**: | | | | |
| Foreign direct investment (FDI) | Foreign direct investment and net outflows. | % of GDP | World Bank online database (2023) | + |
| Trade openness (TRADE) | The sum of exports and imports of goods and services measured. | % of GDP | World Bank online database (2023) | + |
| Government consumption expenditure (GOV_EXP) | Includes all government current expenditures for purchases of goods and services. | % of GDP | World Bank online database (2023) | +/− |
| Gross fixed capital formation (GDI) | Formerly gross domestic investment, which includes land improvements, machinery, and construction of roads. | % of GDP | World Bank online database (2023) | + |
| Population growth rate (POP_GR) | Annual population growth rate for year t is the exponential rate of growth of midyear population from year t−1 to t. | % | World Bank online database (2023) | - |
| Public debt (PD) | Government consolidated gross debt. | % of GDP | Eurostat database (2023) | - |
| Recycle rate (RECYCLE) | Tonnage recycled from municipal waste divided by the total municipal waste arising. | % | Eurostat database (2023) | + |

Table 2 illustrates the statistics for the EU-27 countries and the UK. Over the considered time frame, the lowest value of the dependent variable was USD 5.300 PPS, recorded in 2001 in Romania, whereas the highest was about USD 79.600 PPS in 2019 in Luxemburg.

The average GDP per capita for the sample of EU countries was USD 25.372 PPS. On average, 110.7/100 inhabitants had mobile subscriptions. Notably, the number of broadband subscriptions and internet users is lower compared to mobile phone subscribers. The population grew, on average, by 0.29%, with a minimum of −3.74% and a maximum growth rate of 3.93% in Malta in 2019.

Trade openness records a mean value of 116% of GDP with a maximum of 380% of GDP in Luxemburg in 2019. Foreign direct investments vary greatly across EU countries, with a minimum of −87% of GDP (Malta, 2012) and a maximum of 300% of GDP (Cyprus, 2012). The EU governments spend on average almost 20% of GDP, while the mean value of the gross domestic investment is 22.16% of GDP. Important differences are observed regarding the level of public debt; the average value of public debt in the EU was 59% of GDP, with a minimum level of 3.8% of GDP in Estonia in 2007, and a maximum of 186.4% of GDP in Greece in 2018. Finally, regarding the recycling rate of municipal waste, the average value was 30%, while the top performer in this area was Germany, with a maximum value of 69.7% in 2021.

*4.2. Empirical Model and Method*

The main hypothesis of our empirical study is that digitalisation positively impacts the level of economic growth. The function has the following form:

$$\gamma = f(\lambda) \tag{1}$$

where $\gamma$—economic growth (proxied by GDP per capita); $\lambda$—digitalisation (proxied by the three Information and Communication Technologies variables, i.e., mobile subscriptions, internet users, and fixed broadband subscriptions).

The robustness of the estimation is explored using several econometric models. The extended PLS (panel least squares) naïve panel model is as follows:

$$\gamma_{it} = \alpha + \beta\lambda_{it} + \varepsilon_{it} \tag{2}$$

where $\alpha$—intercept; $\beta$—slope of the interest variable; $i$—country; $t$—time; $\varepsilon_{it}$—the error term, which varies over both country and time.

The impact of digitalisation variables is isolated by entering several control variables that were selected based on data availability and related studies. In this case, the extended linear model becomes as follows:

$$\gamma_{it} = \alpha + \beta\lambda_{it} + \sum_{k=1}^{n} \beta_k X_{k,it} + \mu_i + \eta_t + \varepsilon_{it} \tag{3}$$

where $\alpha$—intercept; $\beta$—coefficient of digitalisation variables; $\beta_k$—coefficient of control independent variable $k$ by $n$ type; $X$—represents the vector of controls, including foreign direct investments, trade openness, government expenditures, gross fixed capital formation, population growth rate, public debt, and recycle rate of municipal waste; $\mu_i$—stands for country fixed effects; $\eta_t$—time-specific effect that controls for unaccounted common time-varying factors; $i$—country; $t$—time; $\varepsilon_{it}$—the random error term.

Firstly, Equation (3) was estimated using PLS regression. Then, as the investigated sample was balanced, we estimated the model in the case of cross-section fixed effects and cross-section random effects. In this context, the Breusch–Pagan test allows a choice between pooled model and fixed-effects model, while the Hausman test indicates the better model between the fixed-effects and random-effects models.

Three scenarios are developed based on the fixed-effects estimator: (1) EU-28; (2) EU-15 (representing the old EU member-states); and (3) EU-13 (EU new member states). This splitting sequence allows us to check for robustness, and also to take into account the fact that EU member states have pursued different economic and social paths in the past, with significant implications on the present approach to economic growth.

## 5. Results

In the first stage of the analysis, the stationarity of the variables was tested to avoid spurious regression using a set of stationarity tests (Levin, Lin, and Chu t*; Im, Pesaran, and Shin W—stat; ADF—Fisher Chi-square; P.P.—Fisher Chi-square). The results illustrate that most of the series are not stationary in level; henceforth, the first differentiation of the series was undergone, with the results indicating that the first-order integrated series are stationary (there was no unit root). Further on, regressions were performed using stationary series. A logarithmic measure for the GDP series was used to reach a smaller amplitude and a better interpretation of the results. No other transformations of the unit of measures were made.

Before performing the panel regression, the multicollinearity in the independent variables was tested, i.e., between the control and the interest variables. To analyse the multicollinearity issue, a matrix of correlations between the individual variables was constructed (Table 4). Based on the results, no multicollinearity issues between independent variables were identified, as all correlation coefficients were lower than 0.8, as indicated by [45].

**Table 4.** Correlation matrix between the independent variables and control variables.

| Correlation | MOBILE | INTERNET | FIXED_SUB | FDI | TRADE | GOV_EXP | GDI | POP_GR | PD | RECYCLE |
|---|---|---|---|---|---|---|---|---|---|---|
| MOBILE | 1.00 | | | | | | | | | |
| INTERNET | 0.26 | 1.00 | | | | | | | | |
| FIXED_SUB | 0.22 | 0.26 | 1.00 | | | | | | | |
| FDI | 0.003 | 0.05 | 0.11 | 1.00 | | | | | | |
| TRADE | 0.08 | 0.04 | 0.07 | 0.15 | 1.00 | | | | | |
| GOV_EXP | 0.11 | 0.02 | 0.06 | 0.001 | 0.44 | 1.00 | | | | |
| GDI | 0.22 | 0.09 | 0.13 | 0.09 | 0.004 | 0.03 | 1.00 | | | |
| POP_GR | 0.05 | 0.01 | 0.05 | 0.22 | 0.08 | 0.14 | 0.05 | 1.00 | | |
| PD | 0.17 | 0.002 | 0.09 | 0.04 | 0.17 | 0.35 | 0.17 | 0.01 | 1.00 | |
| RECYCLE | 0.01 | 0.008 | 0.01 | 0.02 | 0.06 | 0.10 | 0.007 | 0.09 | 0.03 | 1.00 |

Source: own calculation using Eviews 13 software.

The results from Table 4 are confirmed by the variance inflation factor (VIF), which provides a measure of multicollinearity among the independent variables in a multiple regression model (Table 5). A large VIF on an independent variable indicates a highly collinear relationship to the other variables that should be considered or adjusted for in the structure of the model and selection of the independent variables. When the VIF is higher than 10, there is significant multicollinearity that needs to be corrected.

**Table 5.** The variance inflation factor (VIF).

| | Coefficient | Uncentred | Centred |
|---|---|---|---|
| C | $7 \times 10^{-5}$ | 57.2 | |
| MOBILE | $2 \times 10^{-8}$ | 1.47 | 1.22 |
| INTERNET | $1 \times 10^{-7}$ | 2.10 | 1.14 |
| FIXED_SUB | $5 \times 10^{-7}$ | 2.68 | 1.18 |
| FDI | $1 \times 10^{-9}$ | 1.16 | 1.07 |
| TRADE | $2 \times 10^{-8}$ | 1.36 | 1.31 |
| GOV_EXP | $2 \times 10^{-6}$ | 1.46 | 1.44 |
| GDI | $1 \times 10^{-7}$ | 61.39 | 1.24 |
| POP_GR | $7 \times 10^{-6}$ | 1.65 | 1.14 |
| PD | $5 \times 10^{-8}$ | 1.34 | 1.26 |
| RECYCLE | $1 \times 10^{-7}$ | 1.12 | 1.01 |

Source: own calculation using Eviews 13 software.

In the panel-model approach, the model may have heterogeneity in the data. As the investigated sample is balanced, this property was tested in the case of the cross-section fixed-effects model and the cross-section random-effects model (Tables 6–8).

**Table 6.** Empirical results of panel regressions—EU-28 model.

| Dependent Variable: GDP per Capita | | | | |
|---|---|---|---|---|
| **Independent Variables** | **(1)** | **(2)** | **(3)** | **(4)** |
| **constant** | 0.02358 *** (7.810648) | −0.013054 ** (−2.008868) | −0.0160587 * (−1.913904) | −0.013357 * (−2.044835) |
| MOBILE | 0.001949 *** (7.39704) | 0.000877 *** (5.129755) | 0.00069922 *** (4.1704339) | 0.0008437 *** (5.159790) |
| INTERNET | $5.2 \times 10^{-5}$ (0.093684) | 0.000852 ** (2.456660) | 0.0005999 * (1.7906950) | 0.0007901 ** (2.3842214) |
| FIXED_SUB | 0.00174 (1.41736) | 0.000171 (0.221459) | 0.00086178 (1.1247705) | 0.0003318 (0.4464744) |
| FDI | | $-4.5 \times 10^{-5}$ (−1.27514) | −0.00010989 *** (−2.896302) | $-5.6 \times 10^{-5}$ (−1.634578) |
| TRADE | | 0.00077925 *** (4.71909) | 0.00069795 *** (4.358287) | 0.0007635 *** (4.838923) |
| GOV_EXP | | −0.018184 *** (−10.779481) | −0.0180309 *** (−11.068489) | −0.018176 *** (−11.286422) |
| GDI | | 0.0021594 *** (7.5257884) | 0.0023858 *** (6.12524551) | 0.0021837 *** (7.5499910) |
| POP_GR | | −0.0078378 *** (−5.8843741) | −0.01075074 *** (−4.007397) | −0.007872 *** (−5.693486) |
| PD | | −0.00308214 *** (−13.6628) | −0.0031328 *** (−13.87278) | −0.0030903 *** (−14.25112) |
| RECYCLE | | −0.00063124 * (−1.75083) | −0.00083732 ** (−2.361334) | −0.00067444 * (−1.950550) |
| **Type of estimation** | **PLS** | **PLS** | **PLS—FE:CS** | **PLS—RE:CS** |
| **Adjusted R-squared** | 0.10894 | 0.630839 | 0.665260 | 0.6305279 |
| **Durbin–Watson test** | 1.84357 | 1.66825 | 1.930780 | 1.718705 |
| **F-stat** | 22.78139 *** | 93.960990 *** | 30.2200 *** | 93.837126 *** |
| **Akaike info criterion** | −3.39827 | −4.352691 | −4.403373 | |
| **Schwarz criterion** | −3.36748 | −4.265886 | −4.103502 | |
| **Breusch–Pagan test** | | 109.89853 (0.0000) | | |
| **Redundant fixed effects tests** | | | | |
| **Cross-section F** | | | 3.03376503 | |
| | | | (0.0000) | |
| **Cross-section Chi-square** | | | 81.6220226 | |
| | | | (0.0000) | |
| **Hausman test** | | | | 46.463447 (0.0000) |

(...) denotes the t-stat; in the case of the tests, ( ... ) denotes the probability; FE:CS, RE:CS denotes cross-section fixed effects and cross-section random effects; ***, **, and * denote significance at 1, 5, and 10% level of significance, respectively.

**Table 7.** Empirical results of panel regressions—EU-15 model.

| Dependent Variable: GDP per Capita | | | | |
|---|---|---|---|---|
| **Independent Variables** | **(1)** | **(2)** | **(3)** | **(4)** |
| **constant** | 0.0157095 *** (4.6723543) | −0.009538 (−1.339793) | −0.0137674 (−1.473136) | −0.009866 (−1.354424) |
| MOBILE | 0.00111648 *** (3.250573) | 0.00057923 *** (2.985035) | 0.00055321 *** (2.8406230) | 0.0005768 *** (0.0031839) |
| INTERNET | −0.0001462 (−0.253132) | 0.000632342 ** (1.998999) | 0.0005802 * (1.818805) | 0.0006268 ** (1.9820628) |
| FIXED_SUB | 0.0022662 (1.640001) | 0.00126967 (1.648260) | 0.00133696 * (1.710237) | 0.00127191 (1.6503047) |
| FDI | | 0.00010800 (1.4285954) | $7.4 \times 10^{-5}$ (0.894291) | 0.00010698 (1.4040980) |
| TRADE | | 0.00052634 ** (2.506208) | 0.0004578 ** (2.1344654) | 0.00052161 ** (2.4816643) |
| GOV_EXP | | −0.0219635 *** (−10.01548) | −0.0219974 *** (−9.67074) | −0.0219798 *** (−10.002516) |
| GDI | | 0.00144606 *** (4.3453676) | 0.001838 *** (4.064499) | 0.00147145 *** (4.3140909) |
| POP_GR | | −0.0006394 (−0.286471) | −0.0067990 ** (−1.970218) | −0.0008998 (−0.390819) |
| PD | | −0.002226 *** (−9.966545) | −0.002310 ** (−9.51942) | −0.002232 *** (−9.930920) |
| RECYCLE | | −0.0002609 (−0.476032) | −0.0004531 (−0.801621) | −0.0002811 (−0.511645) |
| **Type of estimation** | **PLS** | **PLS** | **PLS—FE:CS** | **PLS—RE:CS** |
| **Adjusted R-squared** | 0.0457337 | 0.6406217 | 0.64154592 | 0.641054 |
| **Durbin–Watson test** | 1.988592 | 1.7444475 | 1.866243 | 1.753442 |
| **F-stat** | 5.856462 *** | 53.051449 *** | 22.775385 *** | 53.14931 *** |
| **Akaike info criterion** | −3.694984 | −4.771323 | −4.867418 | |
| **Schwarz criterion** | −3.646193 | −4.8541504 | −4.553410 | |
| **Breusch–Pagan test** | | 11.077508 (0.0000) | | |
| **Redundant fixed effects tests** | | | | |
| **Cross-section F** | | | 1.0519303 | |
| | | | (0.40216) | |
| **Cross-section Chi-square** | | | 15.673998 | |
| | | | (0.3337) | |
| **Hausman test** | | | | 9.363335 (0.4980) |

(...) denotes the t-stat; in the case of the tests, ( ... ) denotes the probability; FE:CS, RE:CS denotes cross-section fixed effects and cross-section random effects; ***, **, and * denote significance at 1, 5, and 10% level of significance, respectively.

**Table 8.** Empirical results of panel regressions—EU-13 model.

| Dependent Variable: GDP per Capita | | | | |
|---|---|---|---|---|
| **Independent Variables** | **(1)** | **(2)** | **(3)** | **(4)** |
| **constant** | 0.0312952 *** (5.8228287) | −0.0069747 (−0.599406) | −0.0332978 ** (−2.169275) | −0.006974 (−0.61752) |
| MOBILE | 0.0022599 *** (5.730406) | 0.001017 *** (3.759756) | 0.00073544 *** (2.6531896) | 0.0010174 *** (3.873376) |
| INTERNET | −0.0002175 (−0.199575) | 0.00087286 (1.2083754) | 0.00065973 (0.9247633) | 0.0008728 (1.244892) |
| FIXED_SUB | 0.003746 * (1.760305) | −0.0004769 (−0.312655) | −0.00106378 (−0.6848583) | −0.0004769 (−0.322103) |
| FDI | | $-7.5 \times 10^{-5}$ * (−1.685788) | −0.000118 ** (−2.440847) | $-7.5 \times 10^{-5}$ * (−1.7367329) |
| TRADE | | 0.0008138 *** (3.38802) | 0.0008294 *** (3.502716) | 0.0008138 *** (3.4903905) |
| GOV_EXP | | −0.015017 *** (−6.224586) | −0.0150144 *** (−6.380491) | −0.0150178 *** (−6.412693) |
| GDI | | 0.00226069 *** (4.268808) | 0.003539 *** (4.9458465) | 0.0022606 *** (4.3978121) |
| POP_GR | | −0.006748 *** (−3.508735) | −0.010953 *** (−2.6704852) | −0.0067484 *** (−3.61476) |
| PD | | −0.0043505 *** (−10.685083) | −0.0042792 *** (−10.62801) | −0.004350 *** (−11.007986) |
| RECYCLE | | −0.00101197 ** (−2.086956) | −0.0009968 ** (−2.06640) | −0.00101197 ** (−2.150024) |
| **Type of estimation** | **PLS** | **PLS** | **PLS—FE:CS** | **PLS—RE:CS** |
| **Adjusted R-squared** | 0.139952 | 0.646305 | 0.666751 | 0.646305 |
| **Durbin–Watson test** | 1.916945 | 1.791566 | 1.896879 | 1.791566 |
| **F-stat** | 14.9402 *** | 46.865135 *** | 23.82685 *** | 46.865135 *** |
| **Akaike info criterion** | −3.22977 | −4.08197 | −4.097360 | |
| **Schwarz criterion** | −3.17468 | −3.92791 | −3.775229 | |
| **Breusch–Pagan test** | | 56.621257 (0.0000) | | |
| **Redundant fixed effects tests** | | | | |
| **Cross-section F** | | | 2.232177 | |
| | | | (0.0112) | |
| **Cross-section Chi-square** | | | 27.876157 | |
| | | | (0.0058) | |
| **Hausman test** | | | | 26.30116 (0.0034) |

(...) denotes the t-stat; in the case of the tests, ( ... ) denotes the probability; FE:CS, RE:CS denotes cross-section fixed effects and cross-section random effects; ***, **, and * denote significance at 1, 5, and 10% level of significance, respectively.

The Breusch–Pagan test is performed to choose between the PLS model and the fixed effects/random effects (FE/RE) models. The null hypothesis of the BP test is that the PLS model is preferred to the FE/RE models. In our case, for all three scenarios, the probability associated with the test is lower than 10% ($p = 0.000$), which means that the null hypothesis is rejected, and the appropriate model is not the PLS model.

Next, the coefficients for fixed/random effects were estimated to further validate the previous conclusion. Fixed effects indicate individual heterogeneity between countries at the level of the constant, while random effects indicate individual heterogeneity at the level of the entire data set. The redundant fixed effects test was performed, where the null hypothesis states that the PLS model is preferred to the fixed effects model. The probabilities of the F-test and the Chi-square for the cross-section fixed effects reveal (for EU-28 and EU-13) that the null hypothesis can be rejected, which means that the cross-section fixed-effects model is preferred to the PLS estimations. However, in the case of the EU-15 countries, the probability of these tests is higher than the significance threshold (10%), which means that the null hypothesis can be accepted.

Furthermore, the Correlated Random Effects test (Hausman test) was applied to learn whether the random effects estimation is better than the fixed effects estimation. The null hypothesis of the Hausman test is that the RE estimation is preferred to the FE estimation. When looking at the results for the EU-28 and EU-13 countries, we can state that the null hypothesis is rejected (probability is lower than 10%); therefore, the FE estimation is better than the RE estimation. As for the EU-15 countries, the probability associated with the Hausman test is higher than the 0.1 level of significance, which means that the RE estimation is preferred to the FE estimation.

As a general conclusion regarding the choice of the best estimate, based on the performed tests, it can be stated that the fixed-effects model (3) is most adequate to explain the effects that digitalisation (proxied by internet users, mobile subscriptions, and fixed broadband subscriptions) generates on economic growth (GDP per capita) in the case of EU-28 and EU-13 countries. Regarding EU-15 countries, the best estimate is provided by the random-effects model (4). This conclusion is reinforced by the values of the Akaike info criterion and Schwarz criterion. These are often used to choose between competing models. The lower the values of these criteria, the better the model estimation.

Regarding the residual's autocorrelation issue, the Durbin–Watson test value of 2 or nearly 2 indicates that there is no first-order autocorrelation. An acceptable range is 1.70–2.30. After performing the estimation, we can conclude that in both the fixed-effects model (3) and the random-effects model (4), there is no autocorrelation in residuals. Furthermore, we analysed the F-statistic, which captures whether the explanatory variable, digitalisation, is significant in explaining the outcome variable, economic growth. The value of the F-stat and the associated probability (0.000) suggest that the estimations of the chosen models (3) and (4) are correct and statistically significant.

The results of the estimated PLS, FE, and RE models are presented in Tables 6–8. The impact of the independent variables (accounting for digitalisation) and control variables on the dependent variable (economic growth) was examined for all three scenarios.

Regarding the impact of digitalisation on economic growth, the results (Tables 6–8) show that at least one of the interest variables appears significant in all four models and for all three scenarios, and it is positively correlated with economic growth as the dependent variable. In order to increase the robustness of the results, we performed the estimations using an alternate dependent variable, i.e., real GDP per capita growth rate. In this case, the results are similar, meaning that we find a significant positive effect of digitalisation on economic growth (these findings are not included in the tables).

Regarding the impact of digitalisation on economic growth in the EU-28, the results that have been reached after the statistical testing (Table 6) show that two of the three interest variables (i.e., mobile subscriptions and internet users) appear significant in models (2)–(4) and are positively correlated with the GDP per capita. When looking at the control variables, according to the selected FE-PLS model (3), trade openness and gross domestic investment are positively correlated with economic growth. In contrast, government expenditures, FDI, population growth rate, public debt, and municipal waste recycling rate lower the level of the dependent variable.

As for the EU-15 countries, the results from Table 7 are similar to the previous ones, i.e., mobile subscriptions and internet users appear significant in models (2)–(4) and are

positively correlated with GDP per capita. Regarding the control variables, according to the selected RE-PLS model (4), trade openness and gross domestic investment generate a positive impact on economic growth, while government expenditures and public debt record a negative coefficient. The recycling control variables, i.e., FDI, population growth rate, and municipal waste recycling rate are insignificant.

Finally, regarding the EU-13 countries, it seems that only one the interest variable is significant, i.e., mobile subscriptions, which is positively correlated with the GDP per capita. We now turn our attention to the control variables: according to the selected FE-PLS model (3), trade openness and gross domestic investment positively impact economic growth. As for the rest of the control variables (government expenditures, FDI, population growth rate, public debt, and municipal waste recycling rate), their coefficients are negative and statistically significant.

In order to further increase the robustness of our estimations, we decided to include the Digital Economy and Society Index (DESI) as the independent variable. The results of the estimations presented in Table 9 reinforce the conclusions of the previous estimates, i.e., a positive and significant impact of digitalisation (proxied by DESI) on economic growth.

**Table 9.** Empirical results of panel regressions (DESI as independent variable).

| Dependent Variable: GDP per Capita | | | | |
|---|---|---|---|---|
| **Independent Variables** | **(1)** | **(2)** | **(3)** | **(4)** |
| **constant** | 9.33908 *** (91.35775) | 8.98256 *** (52.13374) | 10.15309 ** (91.62625) | 10.00682 *** (102.111) |
| DESI | 0.02335 *** (9.50736) | 0.01162 *** (4.88737) | 0.011397 *** (11.53704) | 0.011412 *** (11.81881) |
| FDI | | 0.001087 * (1.77114) | $3 \times 10^{-5}$ (0.23351) | $5.7 \times 10^{-5}$ (0.448228) |
| TRADE | | 0.00141 *** (4.17800) | 0.00057 (0.96916) | 0.001125 *** (2.73796) |
| GOV_EXP | | −0.00395 (−0.56117) | −0.01183 ** (−2.62346) | −0.01356 *** (−3.35091) |
| GDI | | 0.01104 *** (3.04091) | 0.00024 (0.14794) | 0.000156 (0.09991) |
| POP_GR | | 0.11249 *** (5.22290) | 0.00334 (0.44331) | 0.011354 (1.58392) |
| PD | | 0.00116 ** (2.27236) | −0.00272 *** (−3.55047) | −0.00161 *** (−2.68953) |
| RECYCLE | | 0.00939 ** (7.87750) | 0.000167 (0.21046) | 0.00102 (1.351422) |
| **Type of estimation** | **PLS** | **PLS** | **PLS—FE:CS** | **PLS—RE:CS** |
| **Adjusted R-squared** | 0.40015 | 0.72880 | 0.99136 | 0.57142 |
| **Durbin–Watson test** | 0.06087 | 0.297860 | 1.0896 | 0.77689 |
| **F-stat** | 90.390 *** | 43.6625 *** | 429.690 *** | 22.16603 *** |
| **Akaike info criterion** | 0.24761 | −0.53441 | −3.82134 | |
| **Schwarz criterion** | 0.29065 | −0.33387 | −3.04149 | |
| **Breusch–Pagan test** | | 164.296 (0.0000) | | |

**Table 9.** *Cont.*

| Dependent Variable: GDP per Capita | | | | |
|---|---|---|---|---|
| **Independent Variables** | **(1)** | **(2)** | **(3)** | **(4)** |
| **Redundant fixed effects tests** | | | | |
| **Cross-section F** | | | 140.1171 | |
| | | | (0.0000) | |
| **Cross-section Chi-square** | | | 472.72 | |
| | | | (0.0000) | |
| **Hausman test** | | | | 39.74785 |
| | | | | (0.0000) |

(...) denotes the t-stat; in the case of the tests, ( . . . ) denotes the probability; FE:CS, RE:CS denotes cross-section fixed effects and cross-section random effects; ***, **, and * denote significance at 1, 5, and 10% level of significance, respectively.

## 6. Discussion and Conclusions

The EU countries and the UK are highly heterogeneous regarding GDP per capita, productivity, and digitalisation. As Figure 2 shows, at companies' level, the digitalisation of processes, business, marketing, and environmentally related policies as well as individual digital skills vary widely among these countries. Under these circumstances, the correlation between digitalisation (as the process of data capture and analysis technologies) and economic growth is raised.

As the results of the model show, it can be stated that digitalisation, at large, positively impacts economic growth in the EU-28. The conclusions are placed in a general range regarding the digitalisation trends in the EU-15 and EU-13 groups of countries, without engaging in analysing the specificities of individual countries. This study mainly shows the propensity towards digitalisation in these countries and the differences between them that may explain, at least partly, the discrepancies in economic growth and productivity. By looking closely at the contribution of the selected components of digitalisation in line with [6] (i.e., mobile subscriptions, fixed internet, and broadband subscriptions), conclusions for the EU-15 and EU-13 groups of countries may be nuanced.

The findings are confirmed following the use of the DESI developed by the European Commission to assess the digitalisation progress in the member states. As the results show, the use of the DESI as an independent variable strengthens the robustness of the conclusion that digitalisation positively impacts economic growth, verifying the hypothesis of our research.

As the results of the model show, at the level of the EU-28, digitalisation, in a broad sense, has an overall positive impact in line with the literature [25,28,30,31] when competition, productivity, and aggregate demand increase following its implementation.

Corroborated with the statistics, our results predict that in the EU-15 countries (i.e., the old member states), it is safe to state that the diffusion of digitalisation in companies and institutions is supported by mobile subscriptions and the number of fixed internet users. As expected, trade and domestic investments, such as control variables, positively impact economic growth, whereas environmentally friendly policies that translate into recycling proved to be ineffective. Seemingly, these countries do not rely extensively on FDIs to support growth but rather on domestic capital accumulation and investments. Public spending negatively influences growth, suggesting that either public spending and institutions are not efficient enough and/or they contribute to debt accumulation, which is also confirmed by the negative impact of public debt on growth.

Seemingly, in the EU-13 countries (i.e., the last group that has joined the EU), most digital operations take place using mobile subscriptions. In this group of countries, as opposed to the EU-15 countries, fixed internet is used less, which is apparently because of the difficult accessibility of communication systems in remote or rural areas or the high

subscription costs. It predicts a concentration of digitalised businesses in urban areas (Figure 2a) meaning that digitalisation does not impact growth to its full potential, which is in line with [14,35]. In addition, different ranges of digital skills impact the completion of the process. Moreover, the lack of homogeneity in adopting digital technologies deepens the productivity gap between business sectors, on the one hand, and between countries, on the other (Figure 1b). Although trade openness and domestic investment have the potential to support economic growth, FDIs show a negative impact. In contrast to the EU-15 group, where waste recycling is insignificant, in the EU-13 countries, it negatively influences growth, showing that recycling is not based on innovation and does not induce an added value but rather relies on the overuse of goods. The ratio of recycling shows that environmentally oriented behaviour still needs to be implemented, and arguably, it may hint towards an explanation for the rather low acceptance of digitalisation that embeds the concept of sustainability.

This analysis spans two decades, during which the intensity of digitalisation increased continuously, fostering economic growth and productivity. Nevertheless, to support growth, companies and institutions should consider the new business model elements to increase productivity and aggregate demand in as homogenous a way as possible.

Given the disparities in digitalisation among the EU countries, coordinated policies are needed to harmonise productivity and growth through digitalisation, which is a conclusion that is also supported by [29]. In addition, such dedicated digitalisation policies should be adapted to regional specificities. Indeed, in the case of the EU-27 countries, to be successful, these policies should be adapted to the economic-, cultural-, and digital-related behaviours of each country. Moreover, the public authority could be inspired by the World Bank guidelines [46] concerning the implementation of public policies using ICT.

## 7. Research Limitations

The limitations of the present research, which call for further investigations, refer to the analysis of the digitalisation components (hardware and software) to find which has the highest potential to positively impact growth, productivity, and competitiveness.

More insights are also needed on the other components of digitalisation infrastructure (i.e., the number of computers sold/year, the length of Ethernet installed per capita, etc.), which would provide a clearer view of its contribution to economic growth.

The research does not look into the formation of the digital ecosystem or how the contagion of interconnectivity among domestic and international stakeholders impacts economic growth.

The obstacles and blockages that prevent the extension of digitalisation infrastructure should be studied as support for coordinated policies to eliminate them. As more data become available, an empirical analysis and a comparison between digitalisation policies would be useful to depict which policies are the most effective and could be considered the best practices.

**Author Contributions:** Conceptualization, L.E.D. and P.O.M.; methodology, P.O.M.; software, P.O.M.; validation, L.E.D. and P.O.M.; formal analysis, L.E.D.; investigation, P.O.M.; resources, P.O.M.; data curation, P.O.M.; writing—original draft preparation, L.E.D.; writing—review and editing, L.E.D. and P.O.M.; visualization, L.E.D. All authors have read and agreed to the published version of the manuscript.

**Funding:** This article was supported by the UVT 1000 Develop Fund of the West University of Timisoara.

**Data Availability Statement:** Publicly available datasets were analyzed in this study. This data can be found here: [https://ec.europa.eu/eurostat; https://databank.worldbank.org/; https://digital-agenda-data.eu/datasets/desi/indicators; https://www.itu.int/en/ITU-D/Statistics/Pages/stat/default.aspx] (accessed on 2 March 2023).

**Conflicts of Interest:** The authors declare no conflict of interest. The funders had no role in the design of the study; in the collection, analyses, or interpretation of data; in the writing of the manuscript; or in the decision to publish the results.

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
