# Peer review of "Digitalisation and Economic Growth in the European Union"

_electronics, doi:10.3390/electronics12071718_

Round 1

Reviewer 1 Report

The full scope of digitalisation should be presented: from sensors, monitoring, controling, big data, artificial intelligence as basis for new business motels, digital marketing, communication, etc. It should be described, which of these are analysed in the study; please also explain your selection of proxies in this light

"mobile subscriptions and internet users predict a 540 large diffusion of digitalisation at the companies’ level and in the society" please relate to your analysis

"unless steps are taken by companies 565 and institutions to digitalise their process in order to foster productivity and aggre- 566 gate demand while policies to improve digital skills are delayed," this is common sense, how does this relate to your analysis

for figure 1, it should be described where there are no data

figre 4 should be presented in a legible format

Reviewer 2 Report

Dear authors,

The article is generally easy to follow and requires some style and English corrections. The statistical analysis is well described and supported. However, most of the issues lie in the research design.

In the introduction and literature review, the paper introduces the concept of digitalisation and presents a scattered and superficial review of works related to this concept, with few references from engineering and manufacturing, fields where digitalisation has been researched in depth.

The introduction starts by making general claims about digitalisation without providing references to support such claims. For example, the manuscript states that digitalisation requires the intervention of multiple stakeholders and unleashes creativity. Still, no references are cited supporting this or providing a rationale for such a claim. Also, digitalisation is defined in a general manner. Thus, the scope of the paper is unclear throughout the document. This confusion is evident when digitalisation is defined with its contemporary meaning, which concerns productivity through adopting Industry 4.0 practices and technologies such as The Internet of Things and Artificial Intelligence. However, the study analyses data from two decades ago to infer a correlation between digitalisation and GDP. In the first decade of the sample timeline, AI and IoT were emerging technologies and barely adopted.

Because of a lack of a clear definition of digitalisation, the term is confused with access to particular technologies and infrastructure that enable industry 4.0 practices. In this case, to internet subscriptions. Connectivity to the Internet is not causal for more digitalisation maturity; thus, the study’s conclusions are misleading. Furthermore, the work does not explain how the number of people in a territory that have access to the Internet may reflect the technological capability and practices of the companies that operate in this region.

My biggest concern with this study is the research design to investigate the hypothesis, as evidence is not presented to justify the causal relationship between mobile subscriptions, internet users, and fixed broadband subscriptions to digitalisation.

The methodology and data analysis are well presented and explained, but the conclusions are misleading for the previously mentioned reasons.

I hope my thoughts are well explained and help understand my review.

Reviewer 

Reviewer 3 Report

Review of the article on Digitization and economic growth in the European Union by Petru-Ovidiu Mura and Liliana Eva Donath

The purpose of the article is clearly defined, the authors propose to analyze the impact of digitization on economic growth in the European Union. The issue of impact is ambitious and requires precise measurement of the impact of digitization on EU economic growth. Since the analysis covers a 21-year time period from 2000 to 2021, it is necessary to assume that this impact has a different level, because during this time there were major changes related to the accession of new countries to the EU and the withdrawal of the United Kingdom from the EU.

This problem was dealt with quite well by using models and statistical measures for this purpose.

The weakness of the analysis is that, as the authors point out, there are no scientific studies in the literature that would allow for comparisons of indicators in this area. It should be recognized that the undertaken research will significantly fill the gap in this area. Another weakness is the lack of indications of the impact of digitization on the diverse economic potential of individual EU countries, especially the 13 newly admitted countries.

Hypothesis: The research hypothesis of this paper is that digitization (expressed by fixed broadband subscriptions/100 inhabitants, mobile telephony subscriptions/100 inhabitants and the percentage of Internet users) has a positive impact on economic growth (expressed in GDP/capita) in the EU-27 and the UK Britain. The study covers two decades (from 2000 to 2021) over which digitization took hold.

The hypothesis put forward is interesting, its weakness is that the analysis of the problem was made based on the separation of groups of countries, the so-called of the "old EU", the countries newly admitted to the EU (13), which undoubtedly leads to a high level of generalizations, putting the results of the research into question. The indicators used for the study are widely available in EU statistics, which means that the above analysis should be refined by adopting statistical data for individual countries, which may bring surprising results.

The research results are interesting, but at a high level of generality. As a result, the indicators adopted for the analysis and the research methods used have a methodological dimension, and an empirical dimension is particularly expected, which will allow us to provide recommendations.

Round 2

Reviewer 2 Report

Dear authors, 

You have added new references and revised the writing in some sections to improve the paper’s clarity. 

However, my concerns regarding the research design and conclusions remain the same, and no changes have been made to address these issues. To be able to claim that there is a direct relationship between digitalisation and economic growth with your data, you need to show the causal relationship between the variables selected in your study and digitalisation.

Your introduction must clearly define what you mean by digitalisation, delimiting the project scope. Otherwise, similar approaches to your current study could be, for example, any technology equipment measure as a variable that indicates a country’s digitalisation level. For example, you could then argue that the length of ethernet cable installed per capita indicates a country’s level of digitalisation. Similarly, you could say that the number of computers sold in a country per year per capita does the same. Then you could say that the number of data centres within a country determines its level of digitalisation. But, unfortunately, those claims are assumptions unless the direct causal relationship between those concepts and digitalisation is explained and demonstrated or studies doing it are cited.

Also, there should be a whole section explaining the study’s limitations and to what extent the variables are a measure of digitalisation. Otherwise, the claims mislead readers who can interpret digitalisation in many ways.

In the responses to my previous reviews, you stated that “the proxies for digitalisation were chosen to show the degree of penetration of the internet and mobile phones as prerequisites for digitalisation”. In that case, the study should clearly state that it assesses the correlation between the infrastructure required for digitalisation and economic growth (not digitalisation and economic growth).

Kind regards,

Reviewer 2

Reviewer 3 Report

I propose to publish in a corrected version

Round 3

Reviewer 2 Report

Dear authors,

My previous comments have been mostly addressed, and there is a better explanation of the scope of the findings. The data added to the study helps support your approach and provides more validity. I recommend some minor changes to provide the readers with a clear view of what you define as digitalisation, as currently, many definitions and perspectives from literature are combined. For example, in lines 39 to 41, you state that digitalisation aims to add value to business by “converting information into a digital format”. However, this definition aligns with the concept of ‘digitisation’ and should not be confused with ‘digitalisation’. While the earlier focuses on how the data is captured (analog vs digital), the latter focuses on process transformation.  

Below is an example of how authors from a paper published in the Journal Sustainability define the scope of the term in their study:

“In manufacturing, I4.0 practices include collecting and sharing data using sensors in interconnected devices and using digital technologies for processing and analysing data to take new actions. This article will refer to the process of adopting such data-capturing and analysis technologies as ‘digitalisation’”(Mesa et al., 2022) https://doi.org/https://doi.org/10.3390/su142114358

A specific stance on the definition of digitalisation will avoid misinterpretations of the results presented.

Kind regards,

Reviewer

Author Response

Dear Reviewer 3,

Thank you very much for your kind comments and assistance in improving our article.

  1. I propose to publish in a corrected version.

We have added the definition of digitalisation quoted in the Introduction and made some adjustments to the text (marked blue), and we have included the corresponding reference.

Yours sincerely,